# How Well Do Contemporary Theories Explain Floating Exchange Rate Changes in an Emerging Economy: The Case of EUR/PLN

Adrian Marek Burda

Department of Economics, Cracow University of Economics, 31-510 Krakow, Poland;
adrian.marek.burda@gmail.com

**Abstract:** The purpose of this paper is to investigate how well contemporary exchange rate theories explain fluctuations in exchange rates of emerging economies, before and after the Global Financial Crisis (GFC). As an example, the EUR/PLN exchange rate in 1999–2015 was selected as the currency pair that was the most liquid in the region; it had a stable exchange rate regime in the given period. The whole analysis was performed within the selected linear vector error correction (VEC) model framework. VEC models incorporate such well-known theories as purchasing power parity (PPP), the uncovered interest rate parity (UIP), the Harrod–Balassa–Samuelson (HBS) effect, the terms of trade (TOT), the net financial asset (NFA) theory and risk premium. The results indicate the greater importance of external factors—in particular, the Euro Area (EA) short-term interest rates and EA price shocks after the GFC. The main sources of EUR/PLN variability were found to be exchange rate shocks, terms of trade shocks and foreign and domestic short-term interest rate shocks, as well as foreign price shocks. These results are of particularly high importance for our own exchange rate shocks and indicate that a large part of exchange rate fluctuations in EUR/PLN still remains unexplained.

**Keywords:** exchange rate models; capital-enhanced exchange rate model; behavioral exchange rate model; cointegration





## 1. Introduction

The determinants of exchange rates, both real and nominal, as well as the sources of their variability, are analyzed in many empirical studies concentrating on the period of the Global Financial Crisis (GFC) and the post-GFC period for the Central and Eastern European economies (see, e.g., Kelm 2013; Kębłowski and Welfe 2012; Kębłowski 2015; Boero et al. 2015; Grabowski and Welfe 2020; Dąbrowski and Wróblewska 2016; Dąbrowski et al. 2020; Grabowski and Stawasz-Grabowska 2021; Janus 2020; Kębłowski et al. 2020), as well as for a larger number of developing and emerging countries (e.g., Banerjee and Goyal 2021; Caputo 2018; Edrem and Geyikci 2021).

Most of them pointed out the strong role of the following factors: risk premium—measured by relative credit default swaps (CDS), such as in the works of Kębłowski and Welfe (2012), Kębłowski (2015) and Grabowski and Welfe (2016), or just CDS (Kębłowski et al. 2020); more broadly defined financial shocks (e.g., Dąbrowski and Wróblewska 2016; Dąbrowski et al. 2020); and currency market instability (Grabowski and Welfe 2020). Furthermore, in some of these papers, the significant impact of the terms of trade was indicated (e.g., Kelm 2013; Grabowski and Welfe 2020; Caputo 2018).The role of net financial assets indicated Kębłowski (2015) and Caputo (2018), while the importance of net foreign liabilities different than net foreign direct investments were confirmed in Kelm's (2013) monography. Grabowski and Stawasz-Grabowska (2021) found that the announcement of non-standard monetary policy measures of the European Central Bank (ECB) led to the appreciation of the Polish zloty (PLN), Czech koruna (CZK) and Hungarian forint (HUF) against the Euro

in the period 2010–2019, while Kębłowski et al. (2020) noted that the increase in ECB's balance sheet had a similar effect. Moreover, Edrem and Geyikci (2021) emphasized the important role of both global and regional shocks for Polish currency during the post-GFC period. Kębłowski et al. (2020) also noted the appreciation impact of oil prices on the EUR/PLN exchange rate. On the other hand, the Harrod–Balassa–Samuelson (HBS) effect (Balassa 1964) is found to be insignificant (e.g., Kelm 2013) or is not taken into account in most studies with regard to Central and Eastern European countries. Simultaneously, the significance of the HBS effect is confirmed in a publication by Boero et al. (2015), in which the sample for Poland starts in 1996, and in a publication by Banerjee and Goyal (2021), where the sample includes eight big emerging market economies, such as China, India, Brazil and Russia.

However, most of these studies did not investigate the potential change in relationship between the exchange rate and its determinants over time (before and after the GFC), except for the Kelm monograph (Kelm 2013), whose empirical sections only cover the period up to June 2011. Furthermore, most of the aforementioned studies do not provide impulse response analysis, which is found to be a more appropriate tool for interpreting vector autoregressive (VAR) and vector error correction (VEC) models (Lütkepohl 2005, p. 262) as well as their extensions (such as Markov-Switching VAR—MS-VAR). The only exception is in publications by Dąbrowski and Wróblewska (2016) and Dąbrowski et al. (2020). However, they consider only a relatively parsimonious model, which does not include such effects as the HBS effect or net foreign liabilities different than net foreign direct investments.

In the case of the PLN/EUR exchange rate, we observe a significant appreciation of the Polish zloty in the years 1999–2008, a strong depreciation during the GFC and neither an appreciation nor a depreciation trend in the years 2009–2015. Simultaneously, we observed various economic phenomena, such as a disinflation trend, growing similarity of the Polish and Euro Area (EA) short-term (and to a lesser extent, long-term) interest rates and constant relative productivity growth in the tradable sector in Poland, with fluctuations around the GFC period. Thus, the first research question that we asked was whether the changes in the EUR/PLN trends before and after the GFC were caused by changes in the dynamics of the exchange rate determinants or were due to changes in relationships between them. The second research question was what are the sources of exchange rate variability over different horizons in various periods?

This study contributes to the existing literature in two ways. Firstly, we compared how the impact of economic variables on the exchange rate changed over time, with a focus on the following three periods: before the GFC (up to June 2008), during the Eurozone sovereign debt crisis (up to June 2011) and later (up to December 2015). Secondly, we compared the results using impulse response functions (with bootstrapped confidence bands) and forecast error variance decompositions between different model specifications considered in the literature, namely the purchasing power parity model (PPP), the capital-enhanced equilibrium exchange model (CHEER) and the behavioral equilibrium exchange rate model (BEER), in different variants. Our approach focuses on the significance and direction of the impact of different shocks over specific horizons.

The reasons why we compare the results of such different specifications are (1) the fact that they could be better suited to specific time horizons than others (see the survey in MacDonald 2007) and (2) for the purpose of modelling. The PPP explains the behavior of the exchange rate over very long horizons very well. For shorter periods, BEER (see MacDonald 2007) and CHEER (see Juselius and MacDonald 2004; Kębłowski and Welfe 2010) are found to be more appropriate (Grabowski and Welfe 2020), albeit they are not superior to the PPP model in terms of forecasting, even over short horizons (see Burda 2017).

We focus on the EUR/PLN exchange rate, because it is the most important exchange rate for Central and Eastern Europe. Average daily turnover for exchange rate transactions for PLN was at least 50% higher than HUF and CZK, according to each Survey conducted

in 2001–2019 by Bank for International Settlements (BIS)[1] and about 80% of turnover of spot PLN transactions regarded EUR/PLN.

We apply standard econometric methodology—vector error correction (VEC) model, for I(1) variables only. This framework, which has been widely used in order to estimate equilibrium exchange rate and/or verify several economic hypotheses for the Polish currency, e.g., Rubaszek and Serwa (2009), Kelm (2010, 2013, 2016), Kelm and Bęza-Bojanowska (2005), Kębłowski and Welfe (2010, 2012), Bęza-Bojanowska and MacDonald (2009), Kębłowski (2015) and Kębłowski et al. (2020). We note that some authors applied more sophisticated methodology, such as VEC with I(2) variables (e.g., Kelm 2017); panel VEC—PVEC (Kębłowski 2015; Caputo 2018; Banerjee and Goyal 2021; Edrem and Geyikci 2021), Tobit Cointegrated VAR–Tobit CVAR (e.g., Grabowski and Welfe 2020), Bayesian MS-VAR (e.g., Dąbrowski et al. 2020) or Smooth Transition Autoregressive (STAR) models for modelling residuals dynamics (Boero et al. 2015). However, for the analysed specifications, we found the VEC framework to be sufficiently appropriate.

The structure of the paper is as follows: after the Introduction, there is a brief description of the VECM framework in the second section, which is utilized in this research. The third section presents data and empirical strategy applied to test short-term and long-term determinants of the EUR/PLN exchange. Key stylised facts regarding the EUR/PLN exchange rate and its evolution of potential determinants are depicted in the fourth section. The empirical results, including IRFs and forecast error variance decompositions, are presented and described in the fifth section. The last section presents the conclusions.

## 2. Methodology
### 2.1. Vector Error Correction Model

In the research, standard linear VECMs were utilised, which could be described as follows (Lütkepohl 2005):

$$\Delta Y_t = [Y_{t-1} : D_t^{co}] \begin{bmatrix} \beta \\ \eta \end{bmatrix} \alpha + D_t \xi + \sum_{i=1}^{p-1} \Delta Y_{t-i} \Gamma_i + \sum_{i=1}^{q} X_{t-i} \mathrm{B}_i + \varepsilon_t \qquad (1)$$

where:

$Y_t = (y_{1t}, \ldots, y_{Kt})$ is the vector of $K$ endogenous variables,
$X_t = (x_{1t}, \ldots, x_{Mt}$ is the vector of $K$ exogenous variables,
$D_t^{co}$ includes all deterministic terms in the cointegrated relations,
$D_t$ contains all the remaining deterministic terms (outside the cointegrating relation).

The residual vector is assumed to be a $K$-dimensional, zero mean white noise process with positive definite covariance matrix

$$E(\varepsilon_t \varepsilon_t') = \Sigma_\varepsilon$$

The parameter matrix $\beta$ contains cointegrating relations while matrix $\alpha$—loading coefficients. The matrices $\beta$ and $\alpha'$ have dimensions $(K * r)$ and must have rank $r$, where $r$ is the rank of cointegrating space.

One of the key issues in empirical research is the just selection of the appropriate VECM specification (Lütkepohl 2005). VECMs may include different determinist terms and different endogenous lags, exogenous variables and different cointegration ranks.

In this empirical research in the case of deterministic terms, "case 3" including interception outside the cointegration relationship (Johansen 1991, 2005) and no trends are utilized due to the properties of EUR/PLN and its application as in many other VECM studies for the Polish exchange rate (e.g., Kelm 2013; Rubaszek and Serwa 2009). The lag order selection is typically performed as the first step, so the cointegration rank does not have to be known in order to choose the appropriate lag, while many procedures for specifying the cointegration rank require the selection of lag length. In order to select the optimal lag information criteria (e.g., Akaike 1973; Schwarz 1978), residual autocorrelation (e.g.,

Ljung and Box 1978), Portmanteau tests or Lagrange Multiplier (LM) tests could be used (Lütkepohl 2005, pp. 169–74). However, no single method of the lag selection surpasses the others in terms of detecting the true data-generating process; thus, the researcher should analyse different criteria (Lütkepohl 2005, p. 157).

In order to determine the proper cointegration rank, several tests were proposed, albeit model selection criteria could be used for this purpose (e.g., Lütkepohl and Poskitt 1998). In this research, the *trace test*, *maximum eigenvalue tests* (Johansen 1995, 1988) and the *trace test with a small sample involving the Bartlett correction* (Johansen 2002) were utilized. Ultimately, VECMs were estimated via the well-known maximum likelihood (ML) estimator, called the Johansen procedure (see Johansen 1988, 1991, 1995), which takes account of the rank restrictions for $\Pi = \beta\alpha$.

### 2.2. Structural Analysis—Impulse Response Function and Forecast Variance Error Decomposition

The interpretation of the cointegrated systems should be considered cautiously. Typically, the term from Equation (1) is thought to represent the long-run equilibrium relations between variables. However, this way ignores all other relations between variables which are summarized in the VAR($p$) model or corresponding models (Lütkepohl 2005, p. 262), which may cause that, e.g., $y_2$—innovation on $y_1$, could be quite different from $-\frac{\beta_2}{\beta_1}$ in case of one cointegration relation. Thus, the impulse response may provide a better picture of the relations between variables.

In order to provide the impulse response analysis transformation of VECM (Equation (1)), the following model of its VAR representation is needed:

$$\begin{aligned} A_1 &= \Pi + I_K + \Gamma_1 \\ A_i &= \Gamma_i - \Gamma_{i-1}, i = 2, \ldots, \ p-1 \\ A_p &= -\Gamma_{p-1} \end{aligned} \tag{2}$$

where: $\Pi = \alpha\beta'$.

Thus, the following VAR($p$) process

$$Y_t = A_1 Y_{t-1} + \cdots + A_p Y_{t-p} + \varepsilon_t \tag{3}$$

could be described in such a way that residuals of different equations are uncorrelated. For this purpose, white noise covariance matrix is needed, $\Sigma_\varepsilon = W\Sigma_\varepsilon W'$ which is diagonal matrix with positive element diagonal elements and $W$ is lower triangular matrix with unit diagonal. The decomposition could be obtained from the Choleski decomposition $\Sigma_\varepsilon = PP'$ by defining a diagonal matrix $D$ which has the same main diagonal as $P$ and by specifying $W = PD^{-1}$ and $\Sigma_u = DD'$ (Lütkepohl 2005, p. 58). Thus, orthogonalized shocks are given by $u_t = P^{-1}\varepsilon_t$. Hence, we obtain following impulse response matrices:

$$\Psi_i = \Phi_i P \tag{4}$$

where:

$\Phi_i = \sum_{j=1}^{s} \phi_{s-j} A_j$ for $s = 1, 2, \ldots,$

with $\phi_0 = I_K$ and $A_j = 0$ for $j > p$.

Thus, naturally $\Psi_0 = P$ is lower triangular so that the $u$ shock in the first variable may have instantaneous effect on all the variables, whereas a shock in the second variable cannot have instantaneous impact on $Y_{1t}$ but only on other variables and so on. It should be noted that different arrangement of $Y_t$ may produce different impulse responses. However, alternative ways of obtaining structural shocks have been developed (e.g., Lütkepohl 2005, pp. 358–84) and utilised, and methods described in Equation (4) are still worth considering in applications where competing theories are investigated as in this research. Furthermore, those alternative methods are not immune to typical impulse response analysis issues such

as omitting variable bias. In this research in order to minimize impact those issues we present also bootstrapped interval.

Another popular tool for interpreting VAR/VEC models is forecast error variance decomposition (FEVD). By denoting the $i,j$th element of the orthogonalized impulse response coefficient $\Psi_n$ by $\psi_{ij,n}$, the variance in the forecast error $y_{k,T+h} - y_{k,T+h|T}$ is:

$$\sigma_k^2(h) = \sum_{n=0}^{h-1}(\psi_{k1,n}^2 + \cdots + \psi_{kK,n}^2) = \sum_{j=0}^{K}(\psi_{kj,0}^2 + \cdots + \psi_{kj,h-1}^2) \tag{5}$$

The term $(\psi_{kj,0}^2 + \cdots + \psi_{kj,h-1}^2)$ is interpreted as the contribution variable to the step forecast error variance of variable. By dividing the aforementioned terms by $\sigma_k^2(h)$, we obtain the percentage contribution of variable j in the h-step forecast error variance of variable (Lütkepohl 2005, pp. 63–65).

$$\omega_{kj}(h) = \frac{(\psi_{kj,0}^2 + \cdots + \psi_{kj,h-1}^2)}{\sigma_k^2(h)} \tag{6}$$

## 3. Results

### 3.1. Data Description

We used the monthly data of the nominal exchange rate of EUR/PLN (monthly average) and several macroeconomic variables.

The models are estimated using the monthly data from January 1999 to December 2015 for Poland and the Euro Area. Although the Euro Area expanded throughout the research sample, from 12 countries in 1999 to 19 in 2015, in terms of the data availability and clarity of calculations, most of the data for the Euro Area were calculated for 19 countries (EA19). As all the countries that joined the Euro Area after 2002 are relatively small economies, the impact of this simplification should be low, as the time series for EA19 are very highly correlated with the series for the first 12 Euro Area members (EA12). All the data are revised in the series (as of June 2016) and calculated according to the latest methodologies (e.g., EA 2010 in the case of National Account or BPS6 in the case of the balance of payment statistics). Back extending of data and other adjustments are described below. For estimation purposes, all series envisage the interest rates and risk premium indicator to be transformed to indexes, where values are in terms of 2000 = 1 and are later transformed by natural logarithms.

The data regarding the EUR/PLN exchange rate were obtained from Eurostat and involve the average monthly exchange rates calculated from the daily exchange rate by the National Bank of Poland.

The price indexes for the tradable sector are producer price indexes (PPI) for the manufacturing sector and they are provided by the OECD.

Consumer price indexes are Harmonised Indexes of Consumer Prices (HICP) and they are provided by Eurostat.

Short-term interest rates are three-month WIBOR and EURIBOR rates and they are provided by the OECD. They were, in turn, transformed to monthly interest rates, according to the following formula:

$$i_{m,t} = \left(1 + i_{y,t}\right)^{1/12} - 1 \tag{7}$$

where:

$i_{y,t}$—yearly interest rate,
$i_{m,t}$—monthly interest rate.

Long-term interest rates are the average rates of 10-year government bonds and they are provided by the OECD. As the data from the OECD do not include observations for the period from January 1999 to December 2000, this variable has been back extended, utilising Kelm's (2013) dataset, which included an analogous time series.

The indicator of the Harrod–Balassa–Samuelson effect was calculated by the author, according to the following formula:

$$HBS_t = \frac{\frac{GVA_{MA,pl,t}}{EMP_{MA,pl,t}}}{\frac{GVA_{NMA,pl,t}}{EMP_{NMA,pl,t}}} * \frac{\frac{GVA_{NMA,ea,t}}{EMP_{NMA,ea,t}}}{\frac{GVA_{MA,ea,t}}{EMP_{MA,ea,t}}} \tag{8}$$

where:

*EMP*—denotes the total employment in the sector,
*GVA*—refers to the Gross Value Added in the Sector,
*MA*—manufacturing sector,
*NMA*—aggregate of other sectors than manufacturing,
*pl*—Poland,
*ea*—Euro Area.

All the singular factors needed to calculate HBS are gathered from Eurostat. They were disaggregated from the quarterly time series to the monthly series by the method described by Pipień and Roszkowska (2015), with their own modifications. These modifications refer to the use of specific explanatory variables—the monthly indicators of economic activity and employment in sectors of interest and balancing procedure. Furthermore, due to data availability, the HBS indicator for the period 1999:01–1999:12 was back extended using Kelm's data (Kelm 2013).

The relative terms of trade (TOT) indicator data were taken directly from Kelm (2013) until June 2011 and extended until December 2015, using the TOT indicator calculated by the Polish Statistical Office and export and import price data for EA19 outside EA, provided by Eurostat.

The Net Foreign Direct Investment (FDI) is analysed in relative terms, in relation to GDP:

$$fdi_t = (-1) * \frac{net\ direct\ investment(asset - liabilities)_t}{\sum_{i=t-3}^{t} GDP_i} \tag{9}$$

where the *net direct investments* are quarterly time series, denominated in PLN, gathered from the International Investment Position statistics from the National Bank of Poland and GDP is the quarterly GDP at the current prices gathered from Eurostat.

The quarterly time series was disaggregated to the monthly time series by the use of the automatic interpolation procedure in Gretl Software. Due to a change in the Balance of Payment methodology, data consistent with the new methodology (BMP6) have only been available since 2004. Thus, earlier data were back extended using Kelm's data (Kelm 2013).

Other liabilities than direct investment (OFL) were defined as

$$ofl_t = (-1) * \frac{net\ international\ investment\ position_t - net\ direct\ investment_t}{\sum_{i=t-3}^{t} GDP_i} \tag{10}$$

Data sources and adjustment procedures were analogous, as in the case of FDI.

As a measure of risk premium in this research, the CBOE Volatility Index (VIX) was utilized. This daily time series was aggregated to monthly figures by the unweighted average. VIX is based on the S&P 500 Index (SPX), the core index for U.S. equities, and estimates of the expected volatility, which is calculated by averaging the weighted prices of SPX puts and calls over a wide range of strike prices. Despite the fact that, in reality, it could not truly reflect forward-realized volatility (see, e.g., Adhikari and Hilliard (2014)), it is commonly used by market participants to predict future volatility and could reflect global uncertainty. We found this to be an interesting alternative indicator of risk premium to fiscal indicators commonly used in subject-related literature (Bęza-Bojanowska and MacDonald 2009; Kelm 2013) or credit default swap (CDS) differentials (Kębłowski and Welfe 2012; Grabowski and Welfe 2020), in particular as global shocks played a role in determining the Polish exchange rate (Edrem and Geyikci 2021).

The data are named as follows:

- Nominal exchange rate—EUR/PLN (source: National Bank of Poland)
- "*tradable* sector" price index for Poland—PPI in manufacturing (source: OECD)
- "*tradable* sector" price index for the foreign economy (Euro Area)—PPI in manufacturing (source: OECD)
- Consumer price index for Poland—HICP (source: Eurostat)
- Consumer price index for the foreign economy (Euro Area)—HICP (source: Eurostat)
- Short-term domestic interest rate—3-month WIBOR rate (source: OECD)
- Short-term foreign interest rate—3-month EURIBOR rate (source: OECD)
- Long-term domestic interest rate—10-year Polish government bond (source: OECD)
- Long-term foreign interest rate—average 10-year government bond interest rate in EA19 countries (source: OECD),
- Global risk premium indicator—CBOE volatility index (VIX)
- Variable representing the HBS effect indicator (source: personal calculations based on Eurostat)
- Relative terms of trade indicator—based on Eurostat prices for exports and imports for Poland and EA19 (source: Eurostat)
- Net direct investment (liabilities asset) in relation to GDP (source: personal calculations based on the National Bank of Poland and Eurostat)
- Other liabilities' (net international investment position—net direct investment) relation to GDP (source: personal calculations based on the National Bank of Poland and Eurostat)

All variables, except interest rates, were transformed to natural logarithms. Interest rates were transformed from a yearly to a monthly frequency, according to the following formula:

$i_{m,t} = \left(1 + i_{y,t}\right)^{1/12} - 1$, where $i_{y,t}$—yearly interest rate, $i_{m,t}$—monthly interest rate.

The analysed time series, the expected nominal exchange rate, price indices, short-term interest rates and long-term interest rates for EA19 were back extended by the data utilised by Kelm (2013).

*3.2. Empirical Strategy*

Vectors of endogenous variables ($Y$), in particular VEC models, which we estimated, are written as follows:

$Y_t = [s_t \ p_t \ p_t^*]$—for all considered VEC models for PPP variables, hereon referred to as "*PPP*".

$Y_t = [s_t \ p_t \ p_t^* \ i_{s_t} \ i_{s_t}^* \ i_{l_t} \ i_{l_t}^*]$—is consistent with capital-enhanced exchange rate equilibrium model (CHEER) and hereon referred to as "*CHEER*".

$Y_t = [s_t \ p_t \ p_t^* \ i_{s_t} \ i_{s_t}^* \ i_{l_t} \ i_{l_t}^* \ rp_t]$ is CHEER model where risk premium was added, so later this specification and hereon referred to as "CHEER_RP".

$Y_t = [s_t \ cp_t \ cp_t^* \ i_{s_t} \ i_{s_t}^* \ i_{l_t} \ i_{l_t}^* \ rp_t \ hbs_t]$ is the CHEER_RP model with the HBS Indicator; thus, the specification is called "CHEER_HBS_RP".

$Y_t = [s_t \ cp_t \ cp_t^* \ i_{s_t} \ i_{s_t}^* \ i_{l_t} \ i_{l_t}^* \ hbs_t \ rp_t \ tot_t \ fdi_t \ ofl_t]$ is a hybrid model involving CHEER and the behavioural equilibrium exchange rate (BEER) elements, so the specification is called the "CHEER_BEER" model.

We estimate and analysed separate VEC models for three subsamples as follows: 01 m 1999–06 m 2008; 01 m 1999–06 m 2011 and 01 m 1999–12 m 2015. The reason for such a strategy is the fact that in all of those periods, we may observe different trends in our exchange rate and its potential determinants, which we described in Section 4.

As different lag selection criteria suggested, in many cases, either a large number of lags (e.g., 12—maximum tested) or economically and unjustifiably low ones ($p = 1$), we decided to investigate only $p = 2$ or $p = 3$ via Akaike's criterion and sequential likelihood ratio (LR) tests for the hypothesis VAR with two lags against VAR lags with three lags. Those lags are enough to take account of autocorrelation. Furthermore, forecasting properties of models with this many lags that is better than models with a higher number of lags

(see Burda 2017). If both criteria gave different conclusions, we investigated a further two specifications (with 2 and 3 lags) separately.

In order to choose the appropriate cointegration rank, we take account of the Johansen trace test results with the Bartlett Correction (Johansen 2002) for restricted constant (*rc*) and restricted trend (*crt*) and the Johansen trace test (Johansen 1995) with the corrected sample size for unrestricted constant (*uc*).

In the Impulse Response Function (IRF) and Forecast Error Variance Decomposition (FEVD) analysis, we did not change the variable order in vectors and applied the Cholesky decomposition. As a result, orthogonal shocks in exchange rate in all models have an instantaneous impact on all variables, while the shock in the last variable in the vector has an instantaneous impact only on itself. For example, in the BEER model, the shock of other (non-FDI) foreign liabilities stock has an instantaneous impact only on other (non-FDI) foreign liabilities stock.

In order to assess the statistical significance of the impulse response, we use 90% of bootstrapped confidence intervals and concentrate on short-term (3 months), medium-term (18 months) and long-term horizons (60 months). For cases where 90% of confidence intervals for impulse responses reach both positive and negative territory, but where 75–80% of confidence intervals are concentrated only in positive or negative territory, we assess those impulse responses as "*nearly significant*".

### 3.3. Initial Analysis and Hypothesis

The average annual EUR/PLN exchange in 2015 (4.18) was almost similar to the figures in 1999 (4.23), indicating only a marginal 1.1% nominal appreciation in this time. Even if we take into account some differences in price growth (in the manufacturing sector), we observe only a marginal real appreciation (cumulatively by 3.5%—on average, 0.2% annually—see Figure 1). If we look at consumer price indices as a deflator (see Figure 2), we may find meaningful real appreciation (by 18%, 1.0% annually), which is statistically significant, as a positive linear trend for logarithms of real PLN/EUR (deflated HICP) for the period 1999–01 m–2015–12 m is statistically significant at a 0.001 significance level. In subject-related literature, this is explained by the HBS effect (e.g., Egert et al. 2003; Konopczak and Welfe 2017). However, if we only look at the period after the global financial crisis (GFC), no real appreciation trend could be found as there is no significant linear trend for logarithms of real PLN/EUR (deflated CPI) for the post-GFC period, even if we use February 2009, which was the month with the peak of depreciation (EUR/PLN equaled 4.65) as a starting point for observation. On the other hand, for the period 01 m in 1999:02 m in 2009, we observe a statistically significant positive trend for logarithms of real PLN/EUR (deflated HICP).

While analysing the evolution of short-term interest rates in Poland and the Euro Area (see Figure 3), we could observe their growing similarity over time, which could be caused both by the growing similarity of inflation between Poland and the Euro Area, as well as the weakening autonomy of monetary policy induced by the globalization process (Sławiński 2008). Cour-Thimann and Jung (2020) proved that the response to the evolution of FED interest rates played an important role in determining ECB's monetary policy. Goczek and Partyka (2019) indicated that in the long run, domestic interbank rates in seven EEA countries (including Poland) followed Euro interest rates and positive shocks in EURIBOR, positively and significantly (even in the long run), which impacted domestic inter-bank years. In the years 2000–2001, we observed nearly 15 p.p. of nominal (maximal 8 p.p. real) disparity, due to the initially higher inflation in Poland than in the Euro Area and the restrictive monetary policy in Poland. In the subsequent years, both real and nominal disparity diminished substantially. However, even after the GFC local factors, mainly the observable inflation in Poland, still played a role in determining short-term interest rates. For example, the reference rate of NBP increased (to 4.75%) in May 2012, while at the same time, ECB cut its base rate to almost zero, when faced with a debt crisis and recession in the Euro Area. On the other hand, the impact of non-standard ECB policy measures on

short-term interest rates in Poland (at least in terms of volatility) seems to be limited (e.g., Janus 2020).

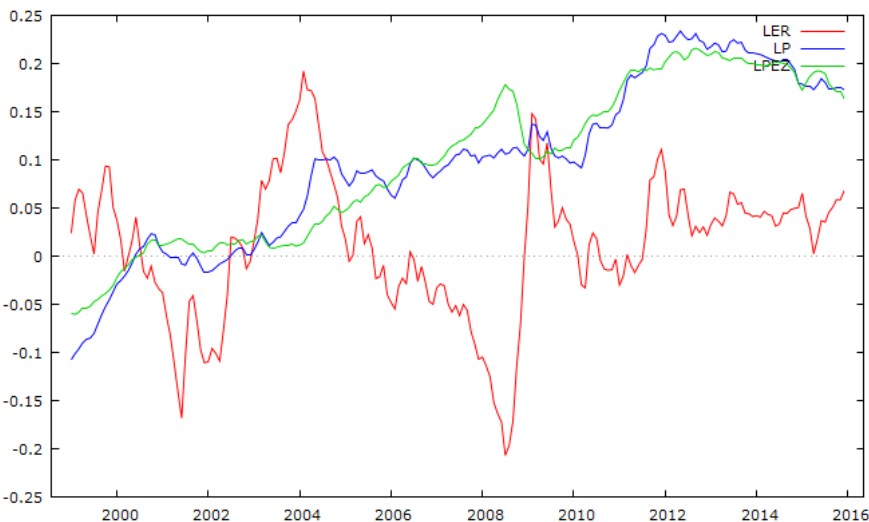

**Figure 1.** Monthly average EUR/PLN exchange rate (LER) and Polish and EA PPI indexes in manufacturing (LP and LPEZ, respectively) between 1 m in 1999 and 12 m in 2015 expressed in logs (see details in Section 3.1).

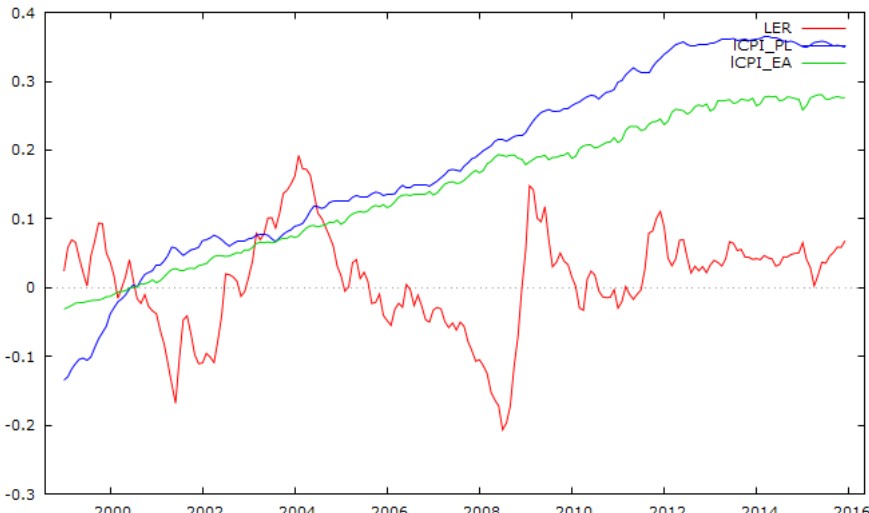

**Figure 2.** Monthly average EUR/PLN exchange rate (LER) and Polish and EA HCPI indexes (lCPI_PL and ICPI_EA, respectively) between 1 m in 1999 and 12 m in 2015 expressed in logs (see details in Section 3.1).

Long-term government interest rates in the analysed period also indicate growing similarity over time (see Figure 4). The largest disparity (reaching around 8 p.p.) was observable in the year 2000, which was caused by inflation disparity between Poland and the EA countries. In the subsequent periods, both nominal and real disparity of long-term interest rates were lower and did not exceed 3 p.p. (2 p.p. after 2009), as they were dependent on inflation expectations and risk premium.

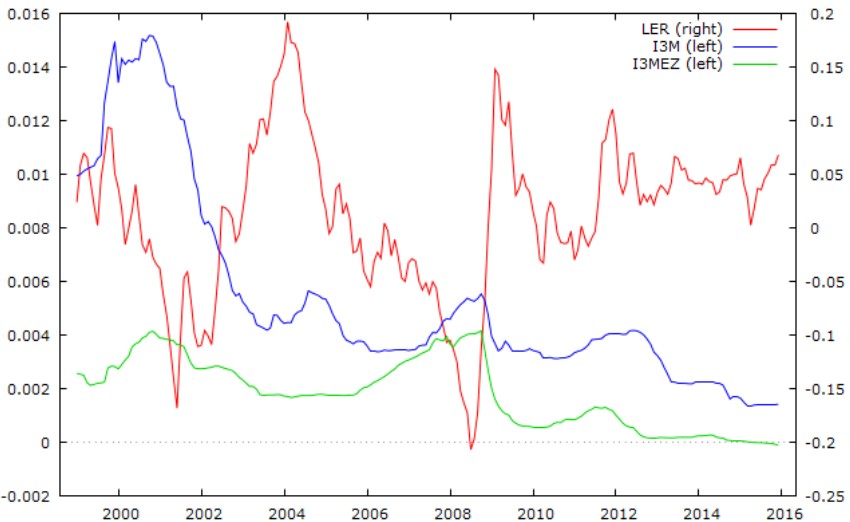

**Figure 3.** Monthly average EUR/PLN exchange rate (LER) expressed in logs and 3 months WIBOR (I3 M) and EURIBOR interest rates (I3 MEZ) expressed in monthly interest rates between 1 m in 1999 and 12 m in 2015 (see details in Section 3.1).

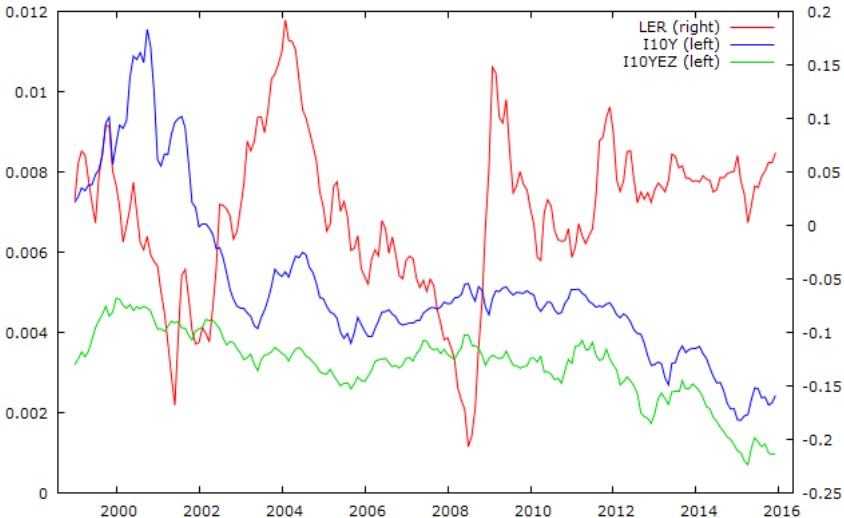

**Figure 4.** Monthly average EUR/PLN exchange rate (LER) expressed in logs and 10-year Polish government bonds (I10Y) and weighted average EA government bond yields (I10 YEZ) expressed in monthly interest rates between 1 m 1999 and 12 m in 2015 (see details in Section 3.1).

CBOE volatility index (VIX) reached its historical maximum during the Global Financial Crisis in October and November 2008, noting higher levels than during other periods with higher global tensions, as in May 2010 or September 2011. Notably, in periods of this spike of VIX, a sizeable depreciation in EUR/PLN was observed (see Figure 5).

Between 2000 and 2015, relative productivity in the tradable sector grew, on average, 2.6% annually (cumulatively 48%), which was much faster than real PLN/EUR (CPI deflated): on average, 1.0% annually between 1999 and 2015[2], which means that the transmission of HBS to prices and real exchange rate was incomplete (see similar findings for the years 1995–2010, in Konopczak (2013)). However, while in the years 2000–2008, relative productivity in the tradable sector in Poland grew, on average, 2.8%, in the years 2009–2015, it grew only by 0.6% on average. Therefore, the obvious question is if we should indicate the slowing convergence after the GFC. A more profound analysis of the data does not suggest such a thesis.

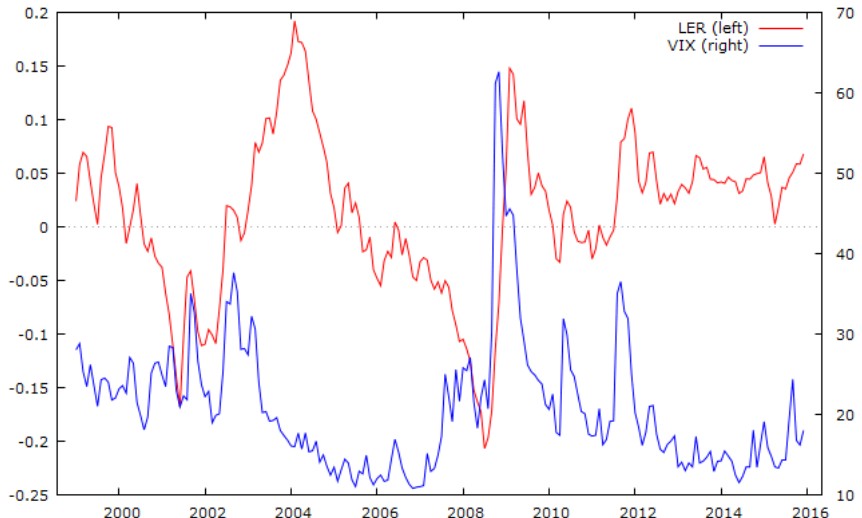

**Figure 5.** Monthly average EUR/PLN exchange rate (LER) expressed in natural logs and VIX index between 1 m in 1999 and 12 m in 2015 (see details in Section 3.1).

Firstly, productivity growth in the manufacturing sector in the years 2009–2015 significantly exceeded the analogous indicator for the Euro Area (5.7% vs. 3.4% annually) in this year (see Table 1). Secondly, during the GFC (between 2008 and 2009), productivity in manufacturing in the EA dropped by over 9%, while in Poland, it grew by over 6%. As a result, relative productivity growth (the HBS indicator) surged to 14%, surging relative productivity growth (the HBS indicator) to 14%. The main cause of this was a deep fall in the GVA in manufacturing in the EA (by 14.5%), while in Poland, the GVA in this sector even showed a slight growth (by 0.7%). Simultaneously, both in Poland and the EA, employment in the manufacturing sector fell to a similar extent (5.2% and 5.6% respectively).

**Table 1.** Productivity changes in *tradable* and *non-tradable* sectors in Poland.

| Period (Annual Data) | CAGR (%) | | | | |
|---|---|---|---|---|---|
| | $\frac{GVA_{MA,pl,t}}{EMP_{MA,pl,t}}$ | $\frac{GVA_{NMA,pl,t}}{EMP_{NMA,pl,t}}$ | $\frac{GVA_{MA,ea,t}}{EMP_{MA,ea,t}}$ | $\frac{GVA_{NMA,ea,t}}{EMP_{NMA,ea,t}}$ | HBS indicator |
| **2000–2015** | 6.6% | 2.2% | 2.0% | 0.4% | 2.6% |
| **2000–2008** | 7.3% | 2.3% | 2.5% | 0.5% | 2.8% |
| **2008–2015** | 5.8% | 2.0% | 1.5% | 0.2% | 2.4% |
| **2008–2009** | 6.2% | 1.6% | −9.4% | −1.2% | 14.0% |
| **2009–2015** | 5.7% | 2.1% | 3.4% | 0.4% | 0.6% |

The relative terms of trade also grew in the period 1999–2015 by 32%, albeit this indicator reached its peak in August 2008—almost at the same time when the "appreciation anomaly" reached its peak (see Figure 6). Kelm (2013, p. 395) observed that in the years 1999–2011, the long-term trend for relative TOT and relative productivity (HBS) was positive, while their deviation from trends was correlated negatively. However, this observation is harder to confirm for the years 2011–2015, when the analysed period was extended to those years (see Figure 7). Similarly, Kelm's conclusion (Kelm 2013, p. 396) that the negative deviation in the relative TOT from its trend had a depreciative impact on EUR/PLN is not so obvious for the post-GFC period (see Figure 8).

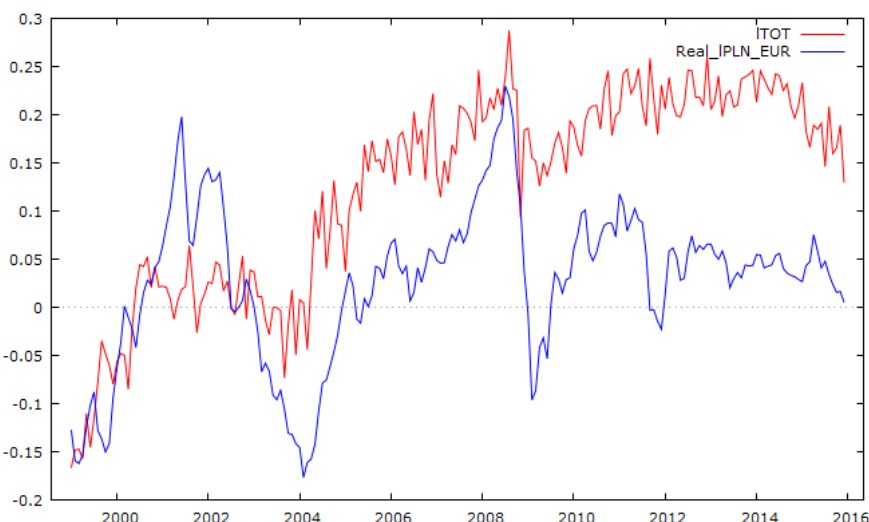

**Figure 6.** Monthly average real PLN/EUR exchange rate (HICP deflated) and terms of trade indicator (lTOT) expressed in natural logs between 1 m in 1999 and 12 m in 2015 (see details in Section 3.1).

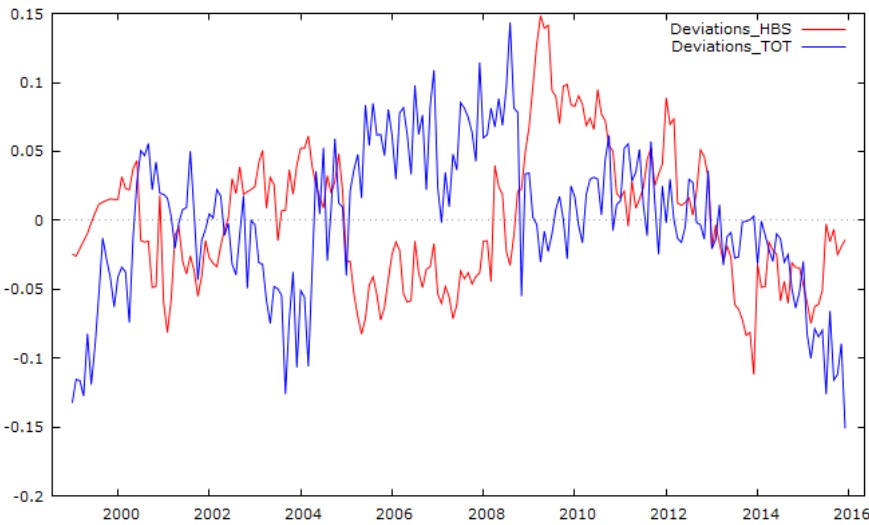

**Figure 7.** Terms of Trade and relative productivity (HBS indicator)—deviations from long-term linear trends between 1 m in 1999 and 12 m in 2015.

In the analysed foreign direct investments, net liabilities increased from 13.4% of GDP in Q1 1999 to 37.5% of GDP in Q3–Q4 2014, declining slightly to 34.5% of GDP in Q4 2015. Simultaneously, other foreign liabilities (net) in the analysed period bottomed out in Q4 2001—9.9% of GDP, yet grew in the subsequent years, reaching peaks (31.8–31.9% of GDP) in Q1 2012 and Q4 2013, while declining to 27.5% in the final years of analysis. While the path of development for FDI is quite stable and nonlinear, the nature of the trend may indicate two phenomena. The first of them is the declining technological gap between Poland and advanced economies. The second is the more complex development of other foreign liabilities, having sizeable deviations from the trend (see Figure 9). Although this could partly reflect the impact of EUR/PLN fluctuations on debt denominated in foreign currencies, we could observe more idiosyncratic movement, such as the increase in OFL around the year 2007 and H1 2008, in spite of PLN/EUR appreciation, as well as the reduction in OFL since 2014, despite the lack of PLN/EUR appreciation (see Figure 10).

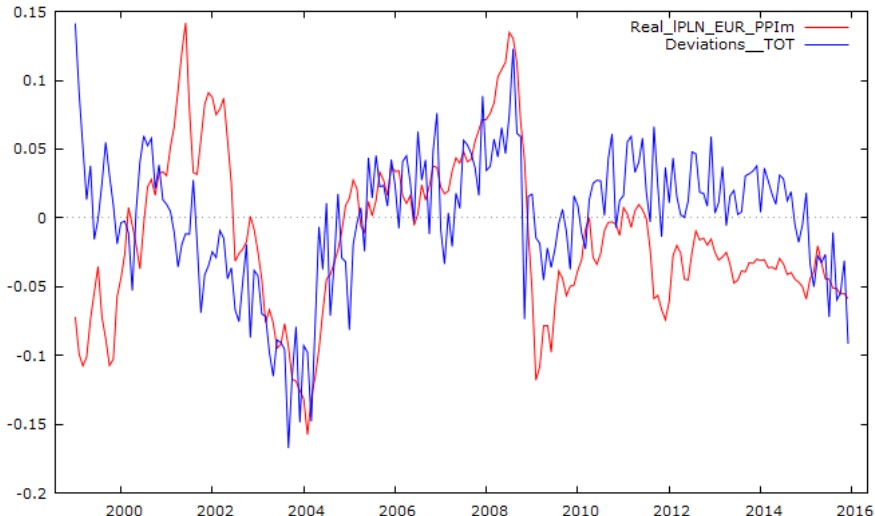

**Figure 8.** Terms of Trade—deviations from long-term trend and real PLN/EUR deflated by PPI in manufacturing expressed in logs—between 1 m in 1999 and 12 m in 2015 (see details in Section 3.1).

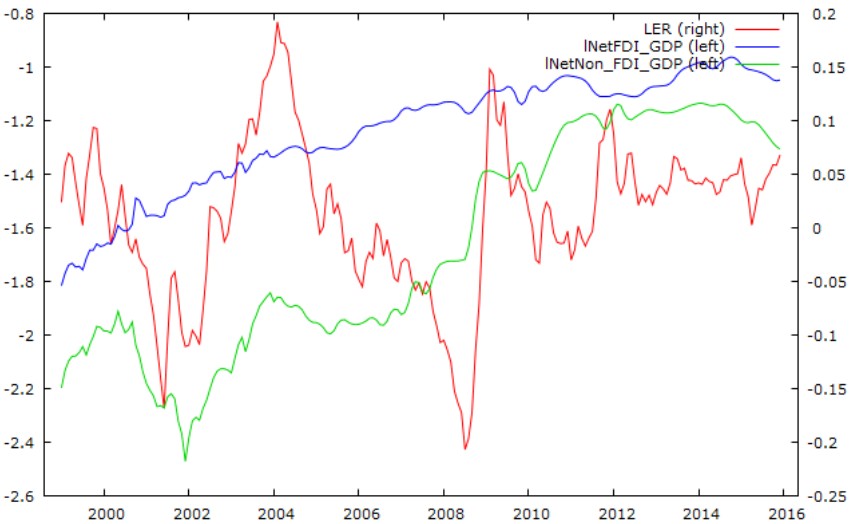

**Figure 9.** Monthly average EUR/PLN exchange rate (LER) and net direct investments on GDP (lNet_FDI_GDP) and other net foreign liabilities in GDP (lNeNon_FDI_GDP) expressed in logs between 1 m in 1999 and 12 m in 2015 (see details in Section 3.1).

### 3.4. Model Selection

In the case of the model with PPP variables, both Akaike's information criterion and LR criterion preferred models with two lags for all subsamples, while for the CHEER_RP model, 3 lags. In other models, these criteria gave different recommendations, so we decided to investigate both specifications (2 and 3 lags).

Not surprisingly, the trace test with the Bartlett correction, as well as the trace tests with only corrected sample size, suggested a different number of cointegration relationships (see Supplementary S1). Thus, we examined the trace test with the Bartlett correction for both restricted trends and the restricted constant and the trace test for the unrestricted constant. We selected the median rank value for those three measures at a 0.1 significance level and provided IRF and FEVD within such a model framework (see Table 2).

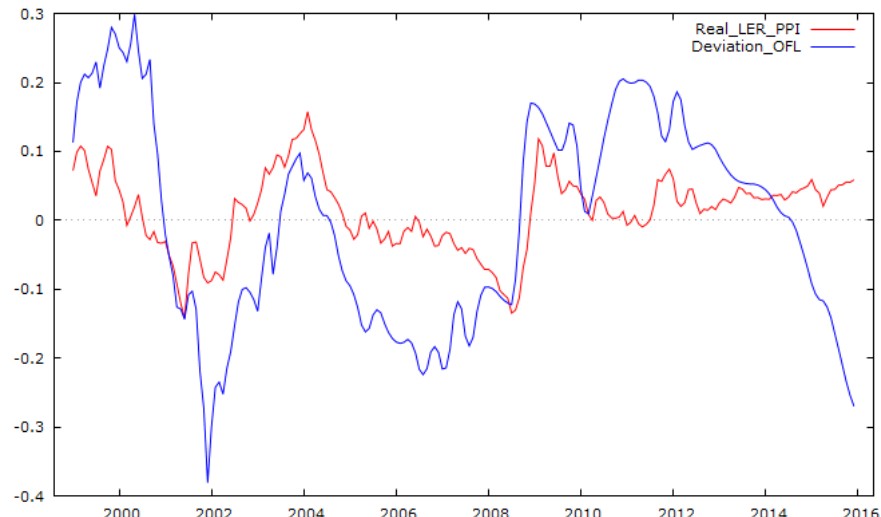

**Figure 10.** Other net foreign liabilities with regard to GDP (OFL)—deviation from the trend and real PLN/EUR deflated by PPI in manufacturing expressed in logs between 1 m in 1999 and 12 m in 2015.

**Table 2.** Selected VEC model specification for further analysis.

| | Preferred Cointegration Rank | | | | | |
| | Sample till 6 m 2008 | | Sample till 6 m 2011 | | Sample till 12 m 2015 | |
| Model | Lags2 | Lags3 | Lags2 | Lags3 | Lags2 | Lags3 |
|---|---|---|---|---|---|---|
| PPP | 1 | | 1 | | 1 | |
| CHEER | | 1 | 1 | 1 | 1 | 1 |
| CHEER_RP | | 3 | | 1 | | 2 |
| CHEER_HBS_RP | 2 | 2 | 3 | 2 | 3 | 2 |
| CHEER_BEER | 3 | 1 | 4 | 4 | 4 | 3 |

### 3.5. Parameter Estimates

We present estimated coefficient of cointegration relationships and adjustment vectors in Supplementary S4. It allowed us to compare models results with results from different papers (as Kelm 2013). Furthermore, in Supplementary S5, we present R-square and adjusted R-square measures for first differences of logarithms of exchange rate. It allows one to assess how the model fits with the data. Adjusted R-square takes into account number of parameters, favouring more parsimonious specifications.

The main conclusions from the analysis cointegration relationship parameter are as follows. Firstly, for PPP models, estimated coefficients for all subsamples suggest that symmetry restrictions occur. Furthermore, adjustment vector in the model estimated for the sample ending in June 2008 shows no error correction for exchange rate deviation (positive coefficient for exchange rate), while for the longer sample, it is negative, indicating error correction and maintaining the PPP relationship. Secondly, parameter estimates for more complicated specifications, even with one cointegrating relationship, such as the CHEER-BEER model with three lags, are ambiguous. Analysis of adjusted R-square measure suggests that for samples ending in June 2008 and December 2015, the CHEER-BEER model with two lags explains the most exchange rate variability, while for the sample ending in June 2015, the CHEER model with three lags fits.

### 3.6. IRF Analysis

The most important synthetic results of IRFs of different shocks on EUR/PLN are presented in Supplementary S3. Not surprisingly, in the case of all model specifications and all subsamples, the impact of EUR/PLN is permanent, almost statistically significant and positive (see Table S1).

Quite surprisingly, the impact of a positive price shock (shock in PPI for PPP, CHEER and CHEER_RP models and shock in HICP for CHEER_HBS_RP and BEER) is negative (meaning appreciation) and statistically or nearly statistically significant for all specifications in the short run. In the medium and long term, this impact is not statistically significant (or at best, nearly statistically significant) in most specifications and the direction of this impulse is not clear (see Table S2).

The impact of foreign price shocks on the exchange rate in Poland according to the investigated specification differs between the pre-GFC subsample and subsamples, including the GFC and the post-GFC period. In the pre-GFC subsample, in most specifications, it is not significant, regardless of the direction, while nearly significant only within the CHEER_RP framework. In subsamples that include the post-GFC period, in particular in subsample 01 m 1999–12 m 2015, the positive foreign price shock drives statistically significant appreciation of EUR/PLN (see Table S3).

The impact of a positive shock in WIBOR3M on the exchange rate for most specifications and subsamples is negative (indicating appreciation), while in most specifications, significant or nearly significant in 1.5-year and 5-year horizons, whereas in the short run, more differentiated in various specifications (see Table S4). Those results are qualitatively similar to the nominal effective exchange rate (NEER) impulse response of monetary policy shocks in the Quarterly Model of (Monetary) Transmission (QMOTR)—where, after initial appreciation, it drops to almost zero after two quarters and partly rebounds in the subsequent quarters (Chmielewski et al. 2018, p. 28).

The impact of a positive EURIBOR3M shock differs in the pre-GFC subsample and subsamples, including the post-GFC period. While in the pre-GFC subsample, the impulse response of EUR/PLN is not statistically significant and has no consistent direction among the specifications, in subsamples, including the post-GFC period, the impulse response is clearly negative and statistically significant or nearly statistically significant (see Table S5). These results could be interpreted in two ways. On the one hand, it could indicate strong spillovers of conventional ECB policy instruments on the Polish exchange market after the GFC. This is partly consistent with the results of a study by Keppel and Prettner (2015)[3], in which they found a statistically significant impact of an increase in the interest rates of the Euro Area on the appreciation of Central European currencies, only in the short run, as opposed to the results of a recent paper by Grabowski and Stawasz-Grabowska (2021)[4], who found no significant spillovers from the ECB's conventional[5] monetary policy measures to the exchange rate markets of the three Central and Eastern European (CEE-3) countries—Poland, Czech Republic and Hungary. On the other hand, in the absence of EA real activity indicators or stock market indicators in the analysed models, EURIBOR3M shocks may partly reflect shocks of real activity, in particular as the biggest ECB interest rate cuts appeared in last months of 2008 and 2011, just when the EA economy entered into recession.

Impact of shocks in 10-year Polish government yields on EUR/PLN exchange rate varies among specification and subsamples, with most models being not statistically significant. However, for the subsample 01 m 1999–12 m 2015, we could observe a "clearly" negative impulse response of EUR/PLN in 1.5-year and 5-year horizons (albeit, in most specifications, not significant), while in a 3-month horizon, positive impulse response (see Table S6).

The impact of positive shock in weighted average yield of 10-year bonds of EA countries for most subsamples and among most specifications is not statistically significant in all horizons (see Table S7).

Contrary to the author's expectations, the impact of the VIX shock on the EUR/PLN exchange rate in subsamples, including the post-GFC period, is not statistically significant. On the other hand, in subsample 1 m in 1999, 12 m in 2015, it is positive (but mainly in 1.5-year and 5-year horizons), which is consistent with the theory. However, for the subsample before the GFC (1 m in 1999, 6 m in 2008), we observe, in many cases (in particular in a month horizon), the negative impulse response of EUR/PLN, which is contrary

to expectations (see Table S8). A graphical analysis of VIX and EUR/PLN trajectories allows us to realize that in late 2007 and early 2008, as in the case of 2001, increasing VIX corresponded with the appreciation of the Zloty (see Figure 5).

The impact of a positive shock in relative productivity (the HBS indicator) on EUR/PLN is not significant within the CHEER_BERR framework when richer structural mechanisms are considered. Interestingly, in the case of the pre-GFC subsample for the CHEER_HBS_RP model, we could observe the impulse response of the EUR/PLN, which is consistent with the graphical analysis performed in Figures 6 and 8. It is also consistent with Kelm's (2013, p. 395) conclusion that the positive deviation in relative productivity induces depreciation and vice versa. However, in subsamples, including the post-GFC period, in particular for subsample 1 m in 1999, 12 m in 2015, this conclusion is not maintained (see Table S9). In our view, other factors that are described below explain the short- and medium-term exchange rate fluctuations better, while the HBS indicator analysis makes more sense in the long- or very-long-term horizon.

The impact of positive shocks of relative terms of trade, FDI to GDP and OFL to GDP on exchange rate was analysed only within CHEER_BEER models. Not surprisingly, the TOT positive shock in all subsamples and horizons induces the appreciation of Zloty against Euro and this effect is particularly visible for subsample 1 m in 1999, 6 m in 2011, which corresponds with Kelm's findings (Kelm 2013, pp. 410–21). However, in subsample 1 m in 1999, 12 m in 2015, its importance, in particular, in a longer-term horizon, is less visible (see Table S10). Interestingly, Grabowski and Welfe (2020), in their model based on sample 2001:01–2018:12, found negative long-term relationships between the EUR/PLN exchange rate and the terms of trade, meaning a significant impact of the increase in the terms of trade on the appreciation of the Polish zloty.

The impact of FDI shock for most specifications and subsamples remains statistically insignificant. Only in the case of sample 1 m in 1999, 12 m in 2015, we observe the expected impact (clearly negative for positive shocks and statistically significant in the long run for some specifications (see Table S10)).

The impact of a positive OFL shock on EUR/PLN in all specifications is consistent with the NFA theory (positive), indicating that the increase in OFL is associated with the depreciation of the Zloty. However, in subsamples, including the aftermath of the GFC period, it is statistically (or nearly statistically) significant only in short-term horizons (see Table S11). All in all, our findings should not be compared directly with the findings of Kelm (2013) or Grabowski and Welfe (2020), due to the different tools used in the structural analysis (interpretation of cointegration matrix vs. IRFs). Furthermore, Grabowski and Welfe (2020) applied overall NFA, similarly to Caputo (2018), without differentiating between FDIs and OFLs. However, our findings are still not contradictory to the aforementioned papers, albeit our analysis suggests that the significance of TOT, FDI and OFLs may be less robust than what the above authors stated.

### 3.7. Main Sources of Exchange Rate Variability—FEVD Analysis

We presented the detailed results for the specific models and subsamples in Supplementary S3, while in this section, we describe the most important findings. Not surprisingly, the exchange rate shocks are the most important source of forecast error variance in the short-term horizons, as they account for 82–98% of forecast error variance, taking account of the positive and significant impact of positive exchange rate shocks on the exchange rate in all specifications for all subsamples. Furthermore, as this is also the case in longer horizons, such as 5 years, it is, depending on the specifications, one of the most important sources of forecast error variance. Relatively speaking, the least important of these shocks is observed for subsample 01 m 1999–06 m 2011. Those results seem to correspond to Dąbrowski et al. (2020), where the financial shocks accounted for over 70% of real exchange rate forecast error variance in turbulent times and 59% in normal times in a one-quarter horizon and this is also an important (in turbulent times, the most important) source of forecast error variance in a 5-year horizon.

The second most important source of exchange rate terms of trade shocks involves foreign price shocks and foreign short-term interest rate shocks, depending on the model specifications. The importance of terms of trade shock are the most robust in subsamples, including the post-GFC period (in particular, in subsample 1999:01–2011:06) in an 18-month horizon. Furthermore, we could observe the growing importance of short-term interest rate shocks and foreign price shocks in the models analysed for the post-GFC subsamples. Foreign price shocks are very important within PPP, CHEER and CHEER_RP, while less important within the CHEER_HBS_RP and CHEER_BEER models, which may suggest the omission of variable bias in specifications without productivity differentials, terms of trade and net foreign asset variables. The importance of foreign short-term interest rate shocks is comparable between specifications, which, in turn, facilitates finding a robust impact. Domestic interest rate shocks have been an important source of medium- and long-term forecast error variance decomposition within some specifications (mostly in the CHEER_RP model analysed for the pre-GFC sample). However, we observe big differences in this shock importance between various specifications (even for the same subsamples). Furthermore, the relative importance of domestic short-term interest rate shocks is lower in the post-GFC subsamples, particularly in subsample 1999:01–2015:12.

## 4. Discussion

In this paper, we investigated determinates of EUR/PLN fluctuations and sources of its variability. We were particularly interested in how those relationships differed across time and different model specifications. In order to investigate them, we estimated various VEC models for subsamples: 1999:01–2008:06; 1999:01–2011:06 and 1999:01–2015:12. Specifications that we considered were: PPP model, CHEER model, CHEER model with risk premium, CHEER model with risk premium and HBS effect and BEER model. For all of them, we applied IRFs and FEVD analysis.

Our results indicate that even within the most complex specifications, exchange rate shocks still played the main role in explaining short- and medium-run exchange rate fluctuations. They have been important also in the long run, having a significant and persistent impact, regardless of the research sample. It may be caused by two types of reasons. The first is the omission of important explanatory variables, which could affect exchange rate. In our opinion, it could be just those components of risk premium, which are not captured by VIX index—mainly including fiscal and domestic policy risk or variables representing demand/output gap or just assessment of instability of currency markets, as proposed by Grabowski and Welfe (2020). The second could be connected with potential non-linear impact of exchange rate deviations—in particular, they could be more persistent if they are large.

Furthermore, terms of trade shocks and foreign and domestic short-term interest rate shocks and foreign price shocks are found to be the next most important sources of exchange rate variability. It is interesting that the results indicate greater importance of external factors—in particular, EA short-term interest rate and EA price shocks after GFC. It has no trivial meaning for economic policy. In particular, it could indicate lowering benefits from independent monetary policy (from ECB) and means that exchange rate could be more responsive to EA demand shocks than domestic shocks.

We see some room for further research, including the following three areas. First, allowance for non-linearities, in particular in adjustments to linear equilibrium relationships (e.g., with STR framework). An alternative solution to this would be a model with time-varying coefficients. The second area is the inclusion of other explanatory variables, as well as exploration of other forms of the model. This would involve the construction of a synthetic risk premium indicator, incorporating both market risk measured by VIX and sovereign risk as CDS, as well as an additional measure of instability of currency market in the region or/and inclusion of demand measure, including global/regional output gap proxies. The third area is simply conducting analysis within a Bayesian approach. All three will be subjects of further research.

**Funding:** This research had been financed by the grant 2014/13/N/HS4/03593 (grant name Preludium) funded by National Centre of Science in Poland.

**Supplementary Materials:** The following supporting information can be downloaded at: https://www.mdpi.com/article/10.3390/economies10110282/s1, Supplementary S1: Summary of cointegration test results. Supplementary S2: Summary of forecast error variance decomposition of EUR/PLN exchange rate for selected shocks. Supplementary S3: Summary of Impulse response function within different models. Table S1: Impact of positive EUR/PLN shock to EUR/PLN Exchange Rate. Table S2: Impact of positive domestic price shock of EUR/PLN Exchange Rate. Table S3: Impact of positive foreign price shock to EUR/PLN Exchange Rate. Table S4: Impact of positive shock of 3-month WIBOR interest rate to EUR/PLN Exchange Rate. Table S5: Impact of positive shock of 3-month EURIBOR interest rate to EUR/PLN Exchange Rate. Table S6: Impact of positive shock of Polish 10-year government bond yield to EUR/PLN Exchange Rate. Table S7: Impact of positive shock of 10-year government bond yield (weighted average for EA) to EUR/PLN Exchange Rate. Table S8: Impact of positive shock of VIX to EUR/PLN Exchange Rate. Table S9: Impact of positive shock of VIX to EUR/PLN Exchange Rate. Table S10: Impact of positive shock of FDI stock to EUR/PLN Exchange Rate. Table S11: Impact of positive shock of other (not FDI) foreign liabilities stock to EUR/PLN Exchange Rate. Supplementary S4: Summary of estimation results of cointegration relation and alpha (adjustment vectors). Supplementary S5: $R^2$ for equation for log changes of logarithm for real exchange rate.

**Informed Consent Statement:** Not applicable.

**Data Availability Statement:** Not applicable.

**Conflicts of Interest:** The author declares no conflict of interest.

## Notes

[1] Turnover of over the counter (OTC) foreign exchange instruments, by currency, Net basis, April 1989–2019 daily average, according to the Triennal Central Bank Survey.

[2] Between 2000 and 2015 cumulative real appreciation was only 3.9% (0.2% annually).

[3] However, in Keppel and Prettner (2015) estimation sample includes earlier years than this research (1995–2009) and exchange rate is just average exchange rate index for 5 Central and Eastern European countries: Poland, Czech Republic, Hungary, Slovenia, Slovakia.

[4] However, in Grabowski and Stawasz-Grabowska (2021) paper more-recent and entirely post-GFC sample is used (2010–2019) and results regarding conventional instruments of ECBs monetary policy are found surprising.

[5] Impact of non-conventional ECB instruments (measured by ECB balance sheet) had been investigated in Kębłowski et al. (2020) who found increase of ECB balance sheet shock causes decline of EUR/PLN. However statistical significance of this impact had been not examined.

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
