# Peer review of "How Well Do Contemporary Theories Explain Floating Exchange Rate Changes in an Emerging Economy: The Case of EUR/PLN"

_economies, doi:10.3390/economies10110282_

Round 1

Reviewer 1 Report

The research paper is interesting and quite current. I have some suggestions for authors, such as:

- line 68 var-iables should be replaced with variables.

- lines 103-121 are identical with lines 137-155. I recommend that authors use the text only once so as not to load the reader with redundant information.

- I suggest authors to include footnotes in the text.

Reading the paper I noticed that the experiment stopped in 2015 and I wonder what would be the reason why the authors did not use the data until 2022. I assume that even if there are no changes in 2015-2022, at least it can be predicted based on history the new turbulent zone.

Author Response

Response to Reviewer 1 Comments

Point 1:  Line 68 var-iables should be replaced with variables

Response 1: I corrected this

Point 2: Lines 103-121 are identical with lines 137-155. I recommend that authors use the text only once so as not to load the reader with redundant information

Response 2: I directly corrected them in the text.

Point 3: I suggest authors to include footnotes in the text.

Response 3: I add most of such citations to the text. I did not for 2 cites in section 3.5, because I did not find the way to add them gently into text.

Point 4: Reading the paper I noticed that the experiment stopped in 2015 and I wonder what would be the reason why the authors did not use the data until 2022. I assume that even if there are no changes in 2015-2022, at least it can be predicted based on history the new turbulent zone.

Response 4: I prefer to not update research and research sample to this article. It is very interesting topic for next article. The main reason is the fact, that it some of variables which I used (as HBS indicator, TOT or OFL) I constructed on my own and it was not straightforward. For example, for construction of HBS I need to make monthly estimates of Gross Value Added and Employment in tradable and non-tradable sectors and back-extend data, using Kelm (2013) dataset. 

At the end I would like to mention you, that I make few changes due to remarks of other reviewers. The most important one include:

             Removal of tables from section 3.5, as they were found unclear and put instead references to appendix where I presented graphs of impulse response functions.

             Change section 2.2. I realised that for submission instead of description of IRF and FEVD, I put content of section 2.1. Now the content of all sections is correct.

Furthermore, I did some minor corrections, including grammar corrections.

Author Response

Response to Reviewer 2 Comments

Point 1: It is an interesting study with slightly unclear aims. The title suggests it is about determinants of the exchange rate, but in fact, the study uses VECM and discusses an impact of different shocks on the different variables with a focus on the exchange rate. Therefore, I suggest the authors to seriously

consider changing the title to something that better reflects what the study actually does.

Response 1: I am struggled to find better title for my work, I found it the most appropriate, as assessment of VEC models typically is performed via impulse response analysis.

Point 2: There is a lack of motivation for the study. Why is it important? I was surprised to see no discussion of the fact that Poland is a member of the EU and plans to be a part of the eurozone in the future. I think it is important to consider the study with this lens

Response 2: I am aware that Poland is aware to adopt Euro, as EU member, but deadline is indefinite. Furthermore, since European Debt Crisis this topic was never in the centre of political agenda. Moreover, current government and central bank governor are strong opponents of Euro adoption. There is even no up to date official (affiliated by National Bank of Poland or Polish Government) analysis of cost and benefits of Euro adoption. Thus, this topic is not present in Polish debate, thus I am not sure if it wise to emphasize this thread in the motivation of the paper.

Point 3: I suggest that the study should be more focused. I found that the discussion in the conclusion makes things clear for the reader so I think this clarity should be brought to the introduction and the main body of the paper.

Response 3: I moved relevant fragment to introduction

Point 4: The authors should discuss how the shocks are identified.

Response 4: I put in the text clearer description that I used Cholesky decomposition and shock order is the same as order variables in the vectors.

Point 5: I believe that one of the main issues with this manuscript is the English grammar, therefore, the manuscript should be professionally copy-edited before publication.

Response 5: I tried to correct English grammar on my own. I will be grateful if paper’s language is fine, or still require some revisions. I will be grateful for one of example of unclear or wrong sentence.

Point 6: Please define all abbreviations before you start using them

Response 6: I did

Point 7: I believe Tables 2-12 can go to the appendix, instead the main results could be described in

words and the difference between various models and subsamples could be emphasized

Response 7: I removed those table (another reviewer found it not informative) and put references to figures at the appendix.

Point 8: Use more conventional notation in equation (1).

Response 8: I use notation of Lutkepohl (2005). I will be grateful for more specific guidance, what is incorrect with this notation.

Point 9: Change the columns name in Tables 2-12: 06m 2008 , 06m 2011 and 06m 2005 look

weird, it ing like pre-GFC etc. Also

replace Lag2 and Lag3 with 2 lags and 3 lags if that s what these mean.

Response 9: I corrected

Point 10: Remove repetitions. For instance, lines 106-121 and 140-155 are almost identical and are

located on the same page

Response 10: I corrected

Point 11: Lines 79-84: chapters should be replaced with sections

Response 11: I corrected

At the end I would like to mention you, that I make few changes due to remarks of other reviewers. The most important one include:

             Removal of tables from section 3.5, as they were found unclear and put instead references to appendix where I presented graphs of impulse response functions.

             Change section 2.2. I realised that for submission instead of description of IRF and FEVD, I put content of section 2.1. Now the content of all sections is correct.

Furthermore, I did some minor corrections, including grammar corrections.

Reviewer 3 Report

1. Section 2.1 and 2.2 are identical

2. It is difficult if not impossible to follow the reasoning in the text. I unfortunately wasn't able to understand most of the paper. The english should be improved.

3. I think the impulse response function results should be presented in the main text. The tables with text descriptions are not helpful to understand the results.

4. In the introduction, I thought the paper would provide a ranking of the different models in terms of forecasting error. But the paper doesn't provide numbers on root mean square error by model.

Author Response

Response to Reviewer 3 Comments

Point 1:  Section 2.1 and 2.2 are identical

Response 1: I realised that I did not put proper content of section 2.2. to submission. I corrected this, be removal of section 2.2. and putting the right content for this section (IRF and FEVD methodology description).

Point 2: It is difficult if not impossible to follow the reasoning in the text. I unfortunately wasn't able to understand most of the paper. The english should be improved.

Response 2: I did several corrections of English grammar, hope that now text is more digestible.

Point 3: I think the impulse response function results should be presented in the main text. The tables with text descriptions are not helpful to understand the results.

Response 3: Having also contradictory remark of another reviewer (ask to put tables in that section in appendix) I decided to remove tables from the text and putting references to exact tables in appendix with impulse response function.

Point 4: In the introduction, I thought the paper would provide a ranking of the different models in terms of forecasting error. But the paper doesn't provide numbers on root mean square error by model.

Response 4: I did not understand this remark. I would be grateful for presentation of the fragment of introduction which is so misleading.

Reviewer 4 Report

Referee Report

This article seeks to explain the movements of the Polish zloty (PLN) relative to the euro for the period from 1999 to 2015. It specifically looks at the global financial crisis (GFC) as a structural break and tries to tie in relevant theoretical models including Harrod-Balassa-Samuelson and net foreign investment, among others.

1.      Heavily reliant on acronyms that are not defined.

2.      Issues with consistency of notation of variables when describing the models.

3.      Very poor English throughout the paper.

4.      Line 284: Why 1999-2008, 1999-2011, and 1999-2015 instead of 1999-2008, 2008-2011, and 2011-2015?

a.      What happened in 2011 to cause a structural break?

5.      Are Figures measured in log difference?

6.      Figures are not references in the body of the paper, making it difficult to match them with the economic analysis.

7.      Line 369: What was the deeper look into the data?

8.      Often, units are not clear. Terms of trade is never clearly defined. Graphs switch from EUR/PLN to PLN/EUR without explanation, which adds to the confusion.

9.      Tables should show results instead of just whether result were significant or not. This will allow readers to determine if the results are economically meaningful.

10.   What are the R-squared values from the regressions? How much variance do these models explain?

Author Response

Dear Reviewer number four,

I would like to thank you very much for your review and remarks.

I respond for your remarks below:

  1. Heavily reliant on acronyms that are not defined.

Author: I corrected this

  1. Issues with consistency of notation of variables when describing the models.

Author: I will be grateful for presentation one of example, unfortunately I was not able to find such case.

  1. Very poor English throughout the paper.

Author: I went through the paper and did some corrections. I hope this will help. If not, I will be grateful for example of poor English which should be changed.

  1. Line 284: Why 1999-2008, 1999-2011, and 1999-2015 instead of 1999-2008, 2008-2011, and 2011-2015?

Author: The reason is the fact, that even sample 1999-2015 is quit short for estimation of big Vector Error Correction model based on monthly data

  1. What happened in 2011 to cause a structural break?

Author: I consider subsample ending 2011:06 due to two reasons. Firstly it includes GFC, but not include fully European Debt crisis. Secondly it correspond well with models estimated by Kelm (2013) who did not presented impulse response function analysis for his models.  

  1. Are Figures measured in log difference?

Author: In table descriptions I explain if there in logs or log-differences or another units. I will be grateful for depiction of doubtful example.

  1. Figures are not references in the body of the paper, making it difficult to match them with the economic analysis.

Author: I corrected this

  1. Line 369: What was the deeper look into the data?

Author: I corrected this

  1. Often, units are not clear. Terms of trade is never clearly defined. Graphs switch from EUR/PLN to PLN/EUR without explanation, which adds to the confusion.

Author: Terms of trade was already described in section 3.1.  I will be grateful for information if you need more

  1. Tables should show results instead of just whether result were significant or not. This will allow readers to determine if the results are economically meaningful.

Author: I remove tables from section 3.5. and referred to impulse response function graphs in appendix. Other reviewers have contradictory view on this. Thus, I hope that the change what I did will be clearer.

  1. What are the R-squared values from the regressions? How much variance do these models explain?

Author: I would be grateful if you specify for which regressions – trend regressions for EUR/PLN or error-correction equations.  

Best regards, 

Author of the paper.

Round 2

Reviewer 3 Report

The paper still need significant english improvement, I would strongly suggest a professional copy-editing service.

Author Response

Dear Reviewer, 

The corrected version of text is after professional copy-editing

which had been done by Native English, who works professional proofreader. I have even invoice for his services. 

Best regards,

Reviewer 4 Report

Referee Report II

The authors failed to meaningfully address the concerns presented in the previous referee report. Details are provided below.

1.      There are still many acronyms which are introduced without being defined first. For example, the opening paragraph refers to EUR, PLN, GFC, and EA without any definitions. In addition, claiming “EUR/PLN significantly appreciated” leaves it vague as to which currency got stronger.

2.      Issues with notation. This is minor, but in equation (1), the authors refer to , but define .

3.      This paper has countless English grammar mistakes in each paragraph, which makes it nearly incomprehensible. The authors should hire a professional copy editor to address this issue.

4.      It is still not clear what the units are for each figure. For example, in Figure 1 the authors claim the exchange rate is measured as EUR/PLN expressed in logs; however, after working through the data a little on my own I have determined that they are probably using PLN/EUR expressed in log difference from the mean. Due to this lack of clarity, I still cannot tell if definition of the spot rate in the model is EUR/PLN or PLN/EUR.

5.      I did not ask for the data tables to be deleted, I asked that the numerical results by provided. In addition, the models should have some measure of fit (such a R-square) to show how much of the variation the model can explain. It is my experience that models of exchange rate movements do a very poor job of fitting the data.

6.      The impulse response functions are very small, which make them very difficult to read.
